# Assessment of Aquifer Recharge Potential Using Remote Sensing, GIS and the Analytical Hierarchy Process (AHP) Combined with Hydrochemical and Isotope Data (Tamassari Basin, Burkina Faso)

**Issan Ki [1,2,*], Hedia Chakroun [1], Youssouf Koussoube [2] and Kamel Zouari [3]**

[1] Hydraulics and Environemental Modeling Laboratory, National School of Engineers of Tunis, University of Tunis El Manar, Tunis 1068, Tunisia

[2] Laboratory of Geosciences and Environment/Training and Research Unit in Life and Earth Sciences, University Joseph Ki-ZERBO, Ouagadougou P.O. Box 7021, Burkina Faso

[3] Laboratory of Radio-Analysis and Environment, National School of Engineers of Sfax, University of Sfax, Sfax 3038, Tunisia

* Correspondence: kiissan84@gmail.com; Tel.: +226-70811350

**Abstract:** In the Tamassari basin, the agricultural population is highly dependent on groundwater resources for its socioeconomic development. However, the decrease in rainfall in the region since the late 1960s and the demographic pressure on the land are significantly affecting groundwater recharge. In order to exploit this groundwater sustainably, it is necessary to identify potential recharge areas for a better capitalisation of this resource. The objective of this study is to map the recharge potential of the existing aquifers making use of remote sensing and GIS techniques and to make a validation based on chloride and tritium contents in the borehole water. The processing carried out on the Landsat 5 and Landsat 8 images combined with a digital elevation model (ALOS PALSAR), highlight the lithological, linear and topographical characteristics of the study area. In addition, various supervised classification algorithms were used to produce the most accurate land use map. Field campaigns were conducted to validate the thematic maps resulting from the geospatial data processing and to collect water samples for hydrochemical (chloride) and isotopic analysis (tritium). The analytical hierarchy process (AHP) was used to derive recharge factors weights. The resulting recharge map shows a perfect agreement between the recharge classes derived from spatial modelling and the tritium isotope analyses. This was not the case with the chloride contents, which showed a dispersion over all the recharge areas.

**Keywords:** Tamassari basin; remote sensing and GIS; recharge potential; hydrochemical; isotopic

## 1. Introduction

Burkina Faso is a landlocked country located at about 500 km from the sea (Atlantic Ocean towards the southwest). Most of the water resources come from rainfall, which generates runoff and groundwater recharge [1]. However, the region has experienced a change in its rainfall pattern since the late 1960s. This change is proven by a period of declining cumulative and daily rainfall [2]. During the 1960s, 1970s and 1980s, the annual number of rainy days decreased significantly in several localities in the region. This reduction averaged 14 days. Annual rainfall then increased over the period 1981–2015. During this period, the country recorded an average rainfall increase of about 31.7% in the Sahelian zone and 22.6% in the Sudano–Sahelian zone. Despite this general trend towards increased rainfall, drought has persisted since 2005 in some localities. This persistence reflects a significant spatial variability in regional rainfall [2]. The decrease in rainfall, together with demographic pressure on the land use, have considerably affected groundwater recharge [2]. Indeed, these natural and anthropogenic changes lead to an increasing

depletion on groundwater recharge [3]. This is coupled with overexploitation of groundwater (more than 60,000 boreholes in 2015) caused by high demand from various users [1] and this is likely to contribute to the lowering of the water table and the disruption of its balance. In the Cascades district of Burkina Faso (Figure 1), current water supply through groundwater abstraction is increasingly characterized by a high failure rate, sometimes reaching 20% [4]. One of the reasons for this high failure rate is the poor knowledge capitalization [5], especially on aquifer recharge, which determines the availability of groundwater resources over time. The Tamassari basin, which is the subject of this study, belongs to this region where groundwater is a primary resource for the socioeconomic development of the population (highly agricultural population) [6]. Therefore, a better knowledge of the recharge in this area becomes an imperative for a better sustainable management of this groundwater resource.

A wide variety of approaches to assessing groundwater recharge exist in different contexts around the world, including hydraulic, isotopic, thermal and numerical methods [7,8]. However, the recharge values vary greatly from one method to another in the same area [9]. Apart from the difference in values depending on the method used to assess recharge, some gaps are due to the fact that spatial variability is not properly assessed because of the low density of the measurement networks reaching 6 km for boreholes or piezometers and up to 50 km for rainfall stations. The spatial extent of the study areas linked to the size of the underlying aquifers requires intensive field exploration for an accurate characterization of the geomorphological variability (lithology, fractures and faults, soil). However, these patterns can greatly influence the potential recharge of groundwater. In this respect, remote sensing and GIS techniques have recently attracted the attention of many researchers [10–13]. In fact, approaches based on spatial analysis techniques improve recharge estimation and provide qualitative assessments of its spatial distribution [14].

The methodology is essentially based on the description, classification and integration of factors influencing recharge [10–16]. For example, in Côte d'Ivoire localized in west Africa, [15] used remote sensing and GIS to identify potential recharge areas of fractured aquifers in the N'zo basin. The results indicate that the potential high recharge areas represent about 20% of the total area of the basin. In north Africa, authors of [16] used the same approaches to map potential recharge areas in the Haouz plain of Morocco. They classified the study area into three descriptive levels of recharge with a range of values from 3.5 to 19%. Furthermore, the recharge potential map using remote sensing and GIS needs to be validated by field data. Thus, isotopic and hydrochemical methods and numerical tools, in particular the calibration of a hydrological model (piezometry, water balance) with available observations can be used as a validation approach [15]. However, the application of numerical methods is very limited due to the existence of many uncertainties related to the conceptualization of the natural environment and the data used in the calibration [17]. The hydrochemical method was applied by [16] in Morocco to validate the recharge map produced from spatial techniques. The hydrochemical tracer used is chloride (Cl-) and the procedure is based on the link between the concentration of chlorides in groundwater and rainwater with annual rainfall. The result of the recharge rate obtained by this approach in the form of a map made it possible to assess with some precision the potential recharge areas of the Houz plain. In the Red Delta plain of Vietnam, the recharge potential map made by spatial techniques was verified by the analysis of the radioactive isotope of the water molecule, tritium ($^3$H) [18]. The different recharge zones were delineated with good agreement with the direct estimation of groundwater recharge by radioactive analysis. The objective of this study is to determine the recharge potential of the aquifers in the Tamassari basin by spatial techniques and its validation using groundwater chloride (Cl$^-$) and tritium ($^3$H) contents. This approach should allow us to judge the relevance of the spatial modelling making use of remote sensing products and GIS integrating data in the assessment of recharge areas based on hydrochemical and isotopic data for the validation in the context of the Cascade region.

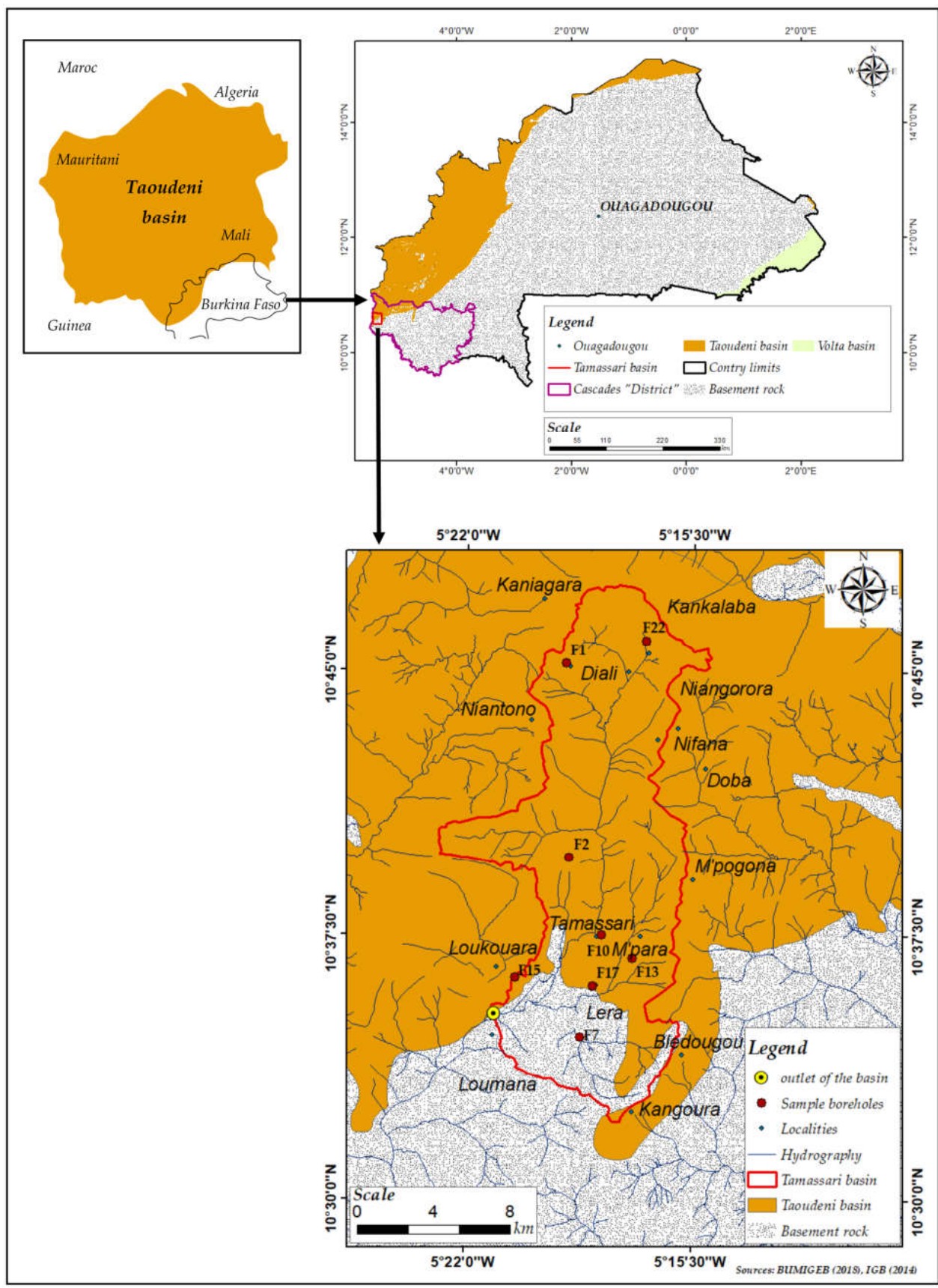

**Figure 1.** Location and geology of the study area.

## 2. Study Area

With an area of 194 km$^2$, the Tamassari basin is located in the southwest of Burkina Faso in the Cascades district, about 515 km from the capital Ouagadougou (Figure 1). It is located between longitudes 5°22′30″ W and 5°15′00″ W and latitudes 10°50′00″ N and 10°50′00″ N.

The Cascades district has a South Sudanese climate marked by two main seasons, a wet season (April to October) with rainfall that can exceed 1400 mm and a dry season (November to March). The average interannual temperatures are between 17 °C and 36 °C [4]. However, it emerges that the rains are often violent and accompanied by devastating winds, making it difficult setting up the crops, causing soil erosion [4]. Potential evapotranspiration values remain very high throughout the year. They are above 100 mm per month. The highest values are observed between February and March, when they reach 200 mm. The lowest values are in July, August and September, when potential evapotranspiration is compensated by rainfall [19]. This situation shows that most of the rain that falls in the study area is consumed by evapotranspiration. In the district, the Fcover indicator shows a progressive degradation of vegetation cover during the period 2007–2014 [20]. The Fcover or fraction of vegetation cover is an indicator that evaluates the proportion of vegetation covering a land surface. In fact, there has been a reduction of 3155 km$^2$ or 16.57% in the cover ratio class between 80 and 100, which constitutes 23.38% of the total area of the district (19,030 km$^2$). At the same time, there was an increase of 787 km$^2$ or 4.14% in the coverage class between 0 and 20, which represents 5.09% of the total area of the district (Figure 2). Concerning the relief, the Tamassari basin has two (02) topographic units, namely the plateaus and the plains, which are crossed by important hydrographic networks. Generally, the dominant soils are ferruginous soils with little leaching and leaching on sandy, sandy clay and clayey materials. Ferralitic soils on sandy-clay are also found in the south and strips of poorly evolved soils on gravel in the central and northern parts of the basin. Hydromorphic soils are poorly represented in the north-eastern part of the basin [21]. These soils are generally poor in fertilising elements, especially nitrogen and phosphorus, with a poor structure. Their physical and hydrodynamic properties are therefore unfavourable for infiltration and water retention [22].

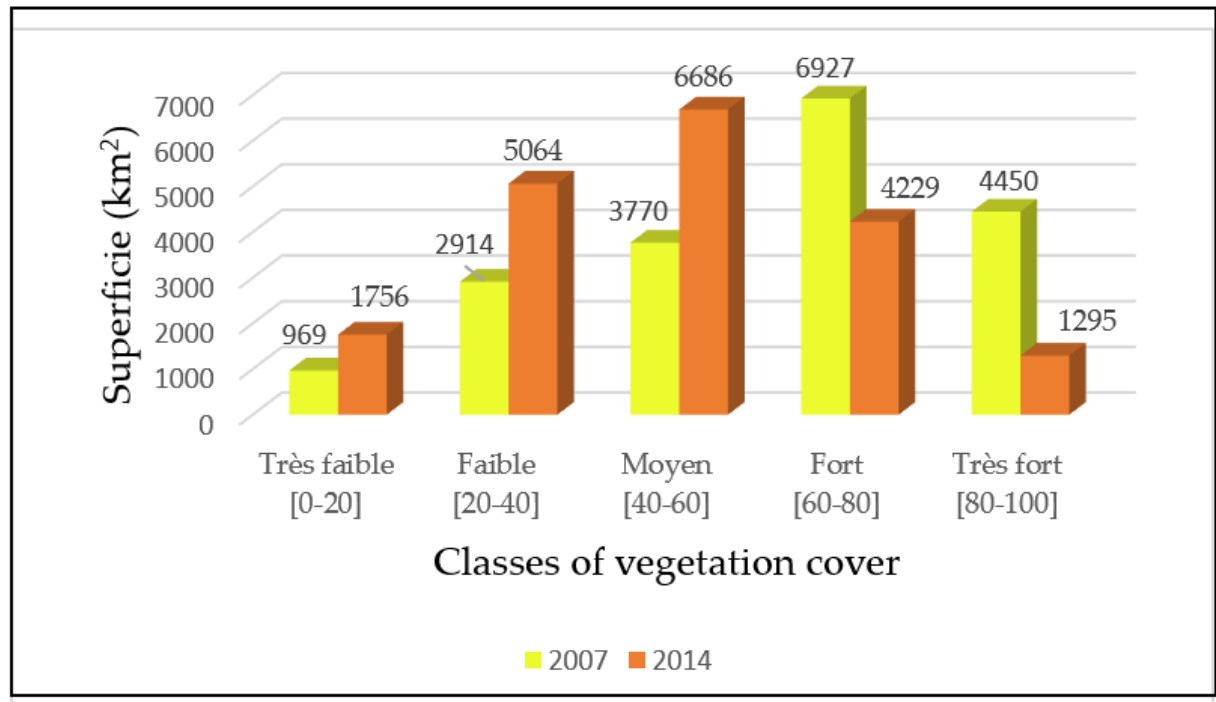

**Figure 2.** Evolution of vegetation cover classes between 2007 and 2014 [20].

From a geological point of view, the study area belongs to the lower siliceous sandstone of Proterozoic age. These are sedimentary formations made up of conglomerate, more or less coarse sandstone with silty layers. These sediments belong to the Taoudéni sedimentary basin, which lies unconformably on the low-permeability basement rock. The domain constituted by the basement rock is characterised by andesites and gneisses dating from the Paleoproterozoic age. On the hydrogeological level, the Taoudéni sedimentary basin constitutes a primordial sandstone aquifer with an enormous capacity to renew its groundwater reserves thanks to the favourable climatological conditions of this region [9,23,24]. However, the piezometric level between 1995 and 2007 showed a downward trend in all seasons despite a slight peak in 2000–2001 following a good rainy season [25] (Figure 3).

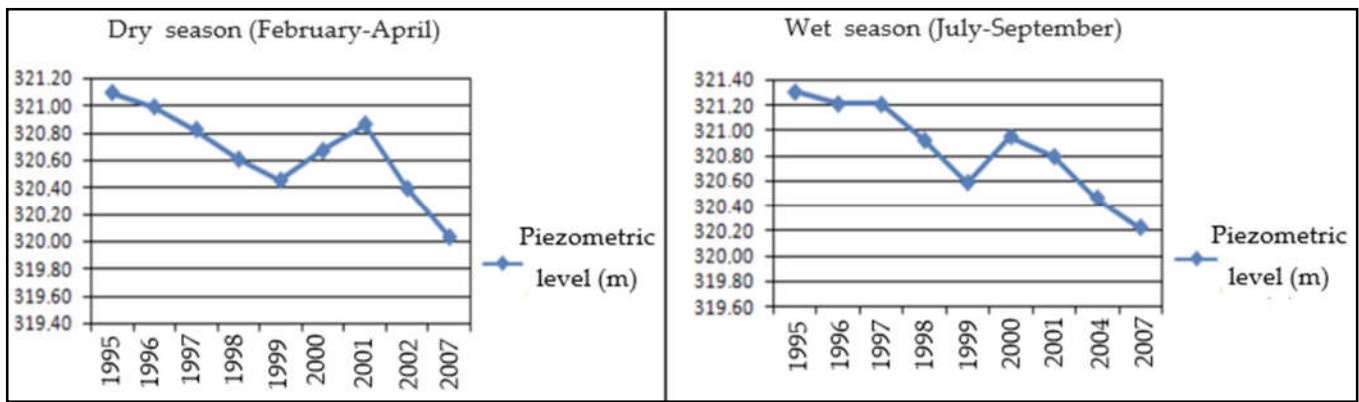

**Figure 3.** Piezometric record between 1995 and 2007 of the Kou and Taoudéni basins [25].

The groundwater flow in this sandstone aquifer is towards the basement rock aquifer [23]. Concerning the basement rock, they form discontinuous aquifers, i.e., fractured aquifers. These fractured basement rock aquifers are sometimes underlying and associated with weathered zones. These weathered zones are 10 m to 80 m thickness and represent the upper part of the basement rock aquifer. The water table in this upper part may be between 5 and 30 m below the ground surface. For fractured basement rock aquifers, their thickness can reach 10 to 80 m and the water table can be between 20 and 60 m below the ground surface [26]. Recharge by rainwater infiltration is generally low [26]. The basin has only one main and temporary river which rises in the commune of Kankalaba, at an altitude of 600 m, and flows from north to south for 28.15 km before joining Loumana downstream at 250 m altitude (Figure 4).

The economic activity is essentially based on the primary sector, particularly agriculture and livestock farming, which employs 91.7% [27] of the active population and contributes with more than 48% to local wealth creation. However, the climatic hazards affecting this sector make it difficult to implement agricultural activities, including livestock farming [4].

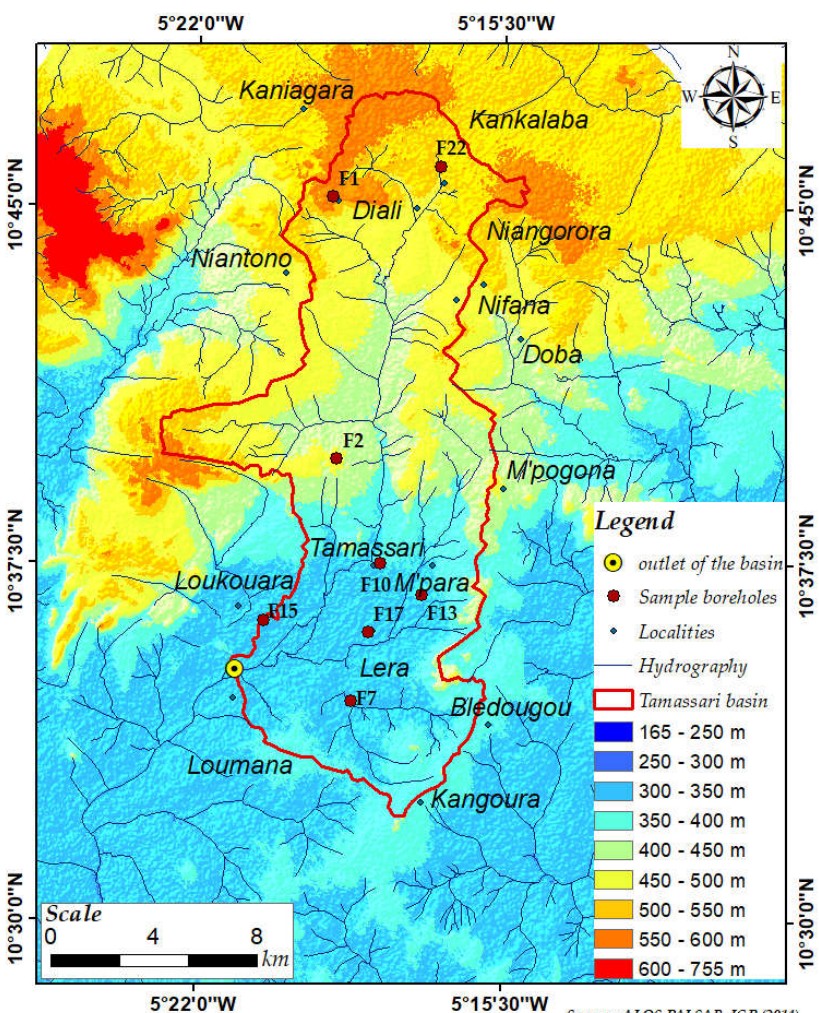

**Figure 4.** Relief of the study area.

## 3. Materials and Methods

The processing of multisource data was used for the extraction of the factors governing aquifer recharge. The classification and crossing of the spatial information of these factors allowed the mapping of the recharge potential of the Tamassari basin. Six steps were important in this process: (i) identification and mapping of factors; (ii) validation of thematic maps; (iii) hydrodynamic classification and standardisation of factor data; (iv) weighting of factors according to the analytical hierarchy process (AHP); (v) mapping of recharge potential; (vi) validation of the recharge potential map. The different steps in this work are represented in the conceptual diagram of the approach (Figure 5).

### 3.1. Data

Several types of geospatial data were used in this study. These were Landsat 5 TM bands 7, 2 and 1 and Landsat 8 OLI bands 5, 4 and 3 acquired free of charge from (glovis.usgs.gov, accessed on 24 August 2022). These 30 m resolution images are identified by their path–row coordinates (197-053) and cover the entire Tamassari basin; they are georeferenced in the UTM WGS 84 (zone 30) coordinate system. Two (02) ALOS PALSAR digital elevation model (DEM) scenes (ALPSRP085610200 and ALPSRP085610210) covering the study area were also downloaded free of charge (asf.alaska.edu, accessed on 24 August 2022) and were also corrected and georeferenced in the UTM WGS 84 (zone 30) coordinate system with a spatial resolution of 12.5 m. As verification tools, national topographic database from the Geographical Institute of Burkina (IGB, 2014) at the scale of 1:200,000 [21] and

geological data at the scale of 1:1,000,000 obtained from the Office of Mines and Geology of Burkina (BUMIGEB, 2018) were required [28]. We also collected field data of structural measurements obtained from outcrops identified in the study area, GPS land use surveys, and water samples taken from eight (8) boreholes during the high-water period (from 15 to 18 July 2021) that corresponds to the period of significant recharge of the water table. Before each water sample was taken in polyethylene bottles, all necessary precautions were taken to avoid any external contamination. The bottles were hermetically sealed, coded and kept in coolers in the dark before their transportation to the chemical and isotopic analysis laboratory. Figure 1 shows the location of the water points sampled in the study area.

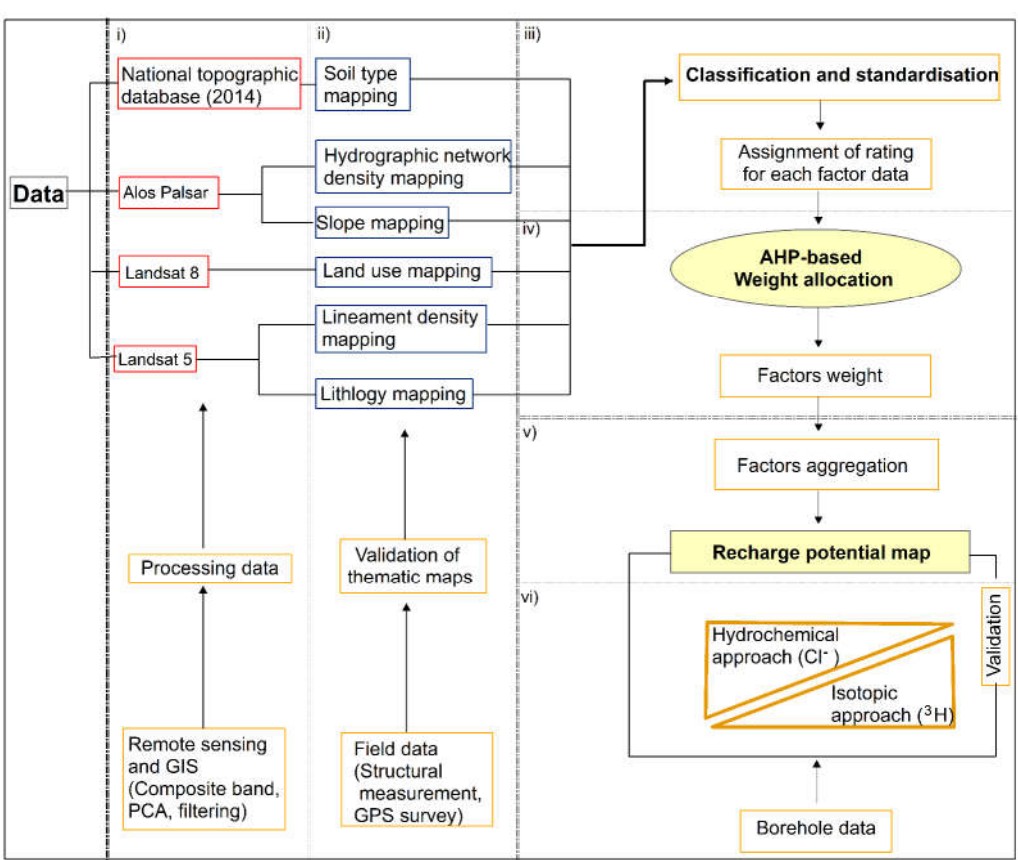

**Figure 5.** Conceptual diagram of the approach.

### 3.2. Spatial Modelling of Recharge

### 3.2.1. Identification and Establishment of Influencing Recharge Factors

The identification of the influencing recharge aquifers factors is a very important phase that conditions the quality of the information to be elaborated. Thus, we refer to hydrogeologic studies on the factors prevailing in aquifer recharge in semiarid regions [11,13–15,29]. Therefore, six (6) parameters were selected in this study due to their importance [29]. These are geology (lithology), soil types, hydrography (drainage density), structure (lineament density), slope and land use. Furthermore, these parameters are physical in nature and change very little over time, unlike climatological parameters that vary from year to year. They can also be easily mapped using remote sensing and GIS techniques.

1. Lithology and lineament density mapping

The 1985 Landsat 5 TM image was used for lithostructural mapping. The choice of this image is explained by the fact that older images are excellent for highlighting lithological signatures and lineaments due to current environmental degradation [30]. However, to highlight the current landscape units of the Tamassari basin, the Landsat 8 OLI image of October 2021 was used. Band 7 of Landsat TM images allows good discrimination between

minerals and rock types [31]. This band is very interesting for the recognition of rocky environments [32]. It has been used in previous studies to map the lithology in Burkina Faso [24,33]. For this reason, several colour combinations were tested with band 7 of the Landsat 5 TM image. The colour composition consists of assigning to each primary colour (red (R), green (G) and blue (B)) three spectral bands acquired by the satellite sensor. Thus, using additive synthesis, it is possible to reconstitute all the colours, thus facilitating the visual image interpretation to highlight environmental phenomena [34]. It is important to note that for each combination of bands, the colour composite images are compared with 1:100,000 scale geological data to ensure that the most accurate lithological map is obtained. In this study, the combination of spectral bands 7, 2 and 1, corresponding, respectively, to the short-wave infrared, green and blue spectral bands of the Landsat 5 TM sensor image acquired in April 1985 showed a good discrimination of the lithological units of the basin. These lithological units were subsequently digitized by visual interpretation on the colour composite image of spectral bands 7, 2 and 1. For lineament density mapping, directional filters were applied to the neo-channels from the principal component analysis (PCA) on bands 7, 2 and 1. Filtering an image allows us to apply a mathematical function to it that modifies the grey values of all or some of the pixels [33]. Thus, in the case of a linear function, this filtering is called linear or directional. The directions concern values from 0° to 315° with a step of 45°. The direction chosen is that of 0° because of its strong contrast recorded on the image. It should be noted that the window size of the filter used is $3 \times 3$, due to the spatial resolution of the image of 30 m. The lineaments were digitized on the filtered image in the 0° direction. For the determination of the lineament density, the lineaments of the basin obtained were discretized into regular grids of $3 \times 3$ km$^2$. In each grid cell, the cumulative length of the lineaments was determined. Interpolation of the cumulative lineament length values per grid cell was used to generate the lineament density map of the basin [15]. Areas with a very high density of lineaments (fractures) are the pole for the existence of a potential reservoir, unlike areas with a very low density of lineaments.

2. Land use mapping

The knowledge of the land use elements provides information on the state of the environment to be crossed by the rainwater, which contributes favourably to the recharge of the aquifers. For the determination of the land cover units of the basin, we applied the colour combination of spectral bands 5, 4 and 3, corresponding, respectively, to the near-infrared, red and green spectral bands of the Landsat 8 OLI sensor image acquired in October 2021. This band's combination is conventionally used in the visual interpretation of land occupation by remotely sensed images [34]. The colour composite of bands 5, 4 and 3 in the RGB channels has the advantage of making visible information that is not visible to the naked eye, and of discriminating well between mineral and vegetation surfaces [34]. The techniques of satellite image classification are very useful for making a land use update map based on training sites defining the main classes and various algorithmic approaches. The training sites correspond to GPS survey fields of land use recognition made during October 2021. We have identified, across the study area, seven (7) classes as those mapped in the region by authors of [21]. We tested the widely used classification algorithms based either on parametric supervised methods (Gaussian mixture model (GMM) [35], K-nearest neighbours (KNN) [35], maximum likelihood [35] or machine learning methods (support vector machines (SVM) [35], random forest (RF) [35]). The classification accuracy assessment is based on the calculation of precision coefficients and kappa index from a confusion matrix comparing algorithms results and field observations of land use.

3. Hydrographic network density and slope mapping

The hydrographic network was generated using ALOS PALSAR DEM data. This product was evaluated for its vertical and horizontal accuracy and had proven to be more suitable for hydrological analysis [36]. The hydrographic network density was calculated as a function of the cumulative length of the hydrographic network on $3 \times 3$ km$^2$ grids as

was carried out previously in the case of the lineament density calculation. Interpolation of the cumulative length values of the hydrographic networks per grid cell was used to generate the density map of the hydrographic networks of the basin [15]. Areas of high hydrographic network density are likely to represent impervious areas where runoff is high. The slope is computed as the maximum elevation gradient of each digital elevation model grid. This factor can provide an idea about recharge areas. Indeed, areas with low slopes can be considered as favourable to infiltration and therefore to recharge. Steep slopes favour rapid runoff and drainage of rainwater. Both hydrographic network and slope result in 12.5 m resolution maps.

4.     Soil type mapping

The soil types of the basin were mapped using existing soil data in the national topographic database of the country [21]. Four soil types were identified in the study area: fersiallitic soils; hydromorphic soils; poorly developed soils; and tropical ferruginous soils. The major effect of soil types on percolation of water into the subsurface media is attributed to its clayey content, as it controls the retention capacity of water [37].

### 3.2.2. Validation of Thematic Maps

The data extracted from the DEM must be verified. Outcrops are the best places to observe and measure structural orientations. To this end, investigations (lithological reconnaissance and structural measurements) were carried out on visible outcrops in the study area. The structural measurements consisted of measuring the direction of fractures observed on an outcrop with a compass. Moreover, observations of some branches of the basin's hydrographic network were made to understand those extracted from the DEM.

### 3.2.3. Hydrodynamic Classification and Data Standardization

Hydrodynamic classification and data normalisation allow the factors to be standardised, as they have been measured at different scales and with different units. The hydrodynamic classification is based on the hydrogeological characteristics of the factor data, in particular their infiltration capacity which favours recharge. To do this, we relied on previous works by [22,38,39]. As an example, for the "lithology" factor, the hydrodynamic classification was based on the flow rate ($m^3$/h) of the aquifers determined by [38] (Table 1). The flow rate of aquifers depends on the presence of fractures (seams, fissures), which condition the water's recharge, but also on the nature of the formations [38]. Thus, the higher the flow rate of an aquifer, the better its hydrodynamic properties (Table 1).

**Table 1.** Influence of geology on the flow rate of aquifers.

| Nature of Rocks | Absolute Maximum Flow Rate ($m^3$/h) | Hydrodynamic Classification |
|---|---|---|
| Sandstone | 12 | Excellent |
| Volcano-sedimentary | 3 | Moderate |
| Shale | 1.6 | Low |
| Green rocks (basalt, andesite, gabbro) | 5 | Average |
| Granite | 6.6 | Good |

Furthermore, it should be noted that the number of hydrodynamic classes has reduced to five (05), as defined in [11,14], to facilitate the multicriteria analysis. These classes are low, moderate, average, good and excellent. Thus, following the same logic, the hydrodynamic classification was made for the other factors, i.e., density of lineaments, land use, density of the hydrographic network, slopes and soil types. Subsequently, a standardised rating scale, ranging from 1 to 10, was assigned to the different hydrodynamic classes following an approach widely used by other authors [11–15] (Table 2).

**Table 2.** Standardised rating scale.

| Hydrodynamic Classification | Low | Moderate | Average | Good | Excellent |
|---|---|---|---|---|---|
| Standardised rating | 1 | 3 | 5 | 7 | 10 |

3.2.4. Factor Weighting with the Analytical Hierarchy Process (AHP)

Weighting is used to assign weights to each factor governing the recharge process regarding its influence. In this study, the weighting method is based on the pairwise comparison by analytical hierarchy process (AHP) developed by authors of [40] and used in various recharge studies [12–14,39]. This method is based on mathematical calculations while assigning weighting coefficients to each factor whose sum is equal to 1.

In order to obtain the weight of the different factors, it is first necessary to set up the comparison matrix. This is obtained by comparing the factors two-by-two on a numerical scale. This matrix, which is noted A, is a square matrix of dimension $K \times K$, where $K$ represents the number of evaluation factors. If we put $a_{ij}$ as a term of A (with $i$ the $i$-th row and $j$ the $j$-th column), then each term $a_{ij}$ represents the relative score of the $i$-th parameter compared to the $j$-th. The comparison matrix takes the form provided by Equation (1).

$$A = \begin{bmatrix} 1 & a_{12} & . & . & . & a_{1K} \\ 1/a_{12} & 1 & . & . & . & a_{2K} \\ . & . & 1 & . & . & . \\ . & . & . & 1 & . & . \\ . & . & . & . & 1 & . \\ 1/a_{1K} & 1/a_{2K} & . & . & . & 1 \end{bmatrix} \tag{1}$$

The aggregation used in the AHP method is based on the calculation of a "weight" vector W from the comparison matrix. The eigenvector of the matrix A is one of the best solutions used to determine the overall assessment of each action. By solving Equation (2), the maximum eigenvalue $\lambda_{max}$ and the eigenvector W are calculated [41].

$$A.W = \lambda_{max}W \tag{2}$$

The $\lambda_{max}$ value is always greater than the rank of the matrix (n). For a matrix close to the consistency condition, $\lambda_{max}$ max is close to (n). Thus, a consistency index CI was defined (Equation (3)) to measure the degree of consistency of the matrix by calculating the product of the comparison matrix by the weight vector W and summing the elements of the resulting vector.

$$CI = \frac{(\lambda_{max} - n)}{(n - 1)} \tag{3}$$

The consistency ratio (CR) of the matrix is defined by Equation (4), which compares the CI to a random index (RI) generated independently of the sample size [40]. CR should not exceed 10% for acceptable consistency.

$$CR = \frac{CI}{RI} \tag{4}$$

3.2.5. Recharge Potential Mapping

The establishment of the synthesis map of the recharge potential consists in transferring in space, the different values of the recharge indices ($R$) intervening in the elaboration of that map. Thus, the calculation of the recharge indices consists of summing the standardised ratings of the different factors multiplied by the normalized eigenvector ($w_i$) of each factor (see Equation (5)).

$$R = \sum_{i=1}^{K} X_i \; x \; w_i \tag{5}$$

where $X_i$ is the standardised rating of the factor $i$.

To facilitate the interpretation of the recharge potential map, authors of [14] defined five (5) recharge areas to which the recharge indices values relate, and, in the same logic, the recharge indices values of the basin were interpreted. These are: high recharge areas, average to high recharge areas, average recharge areas, low recharge areas and very low recharge areas.

### 3.3. Hydrochemical and Isotopic Analysis

To judge the relevance of the applied method, the recharge potential map was compared with borehole data collected during the high-water period from 15 to 18 July 2021. These data include information collected on the water of eight (08) boreholes belonging to the Tamassari basin and concern the chloride and tritium contents (Figure 1). The choice of these two parameters for the validation of the recharge potential map is justified by the fact that they are the most commonly used in arid and semi-arid regions to investigate recharge issues [9,16,29]. Modern recharge can be easily identified from chemical and isotopic signatures (Cl, $\delta^{18}$O and $^3$H) in the unsaturated zone and in shallow aquifers [42]. In many recharge studies, tritium has been used effectively to identify sources of groundwater recharge [1,9,30,43–45]. Indeed, the presence of tritium in the waters indicates a current recharge when the natural contents in the waters (before the thermonuclear tests in 1952) do not exceed the units (1 TU) [44,45]. However, since the thermonuclear tests started in 1952, the tritium content in the waters increased until early 1960s. Its content reached 1111 TU at the IAEA station in Bamako in 1964. Since then, its content has only decreased, reaching a low value of 5 TU in 1998 at the IAEA station in Bamako [43]. In the Sourou basin, a basin close to the study area, the current average tritium content in rainwater is 5 TU in 2010 [30]. We believe that this value has probably decreased today. In all cases, more closely the tritium content of the groundwater is to this value, more important is the recharge. For chloride, several authors have already used it to address various groundwater management issues, including recharge [16]. The principle is based on the fact that the only source of chlorides in the unsaturated zone is rainfall [16]. The higher their concentration in the water table, the higher is the recharge. Thus, the chloride and tritium contents of the collected borehole water were used to validate the different recharge zones mapped using the abovementioned approach.

## 4. Results

### 4.1. Tritium and Chloride Contents in Groundwater in the Basin

#### 4.1.1. Tritium Contents of Groundwater

Tritium contents in recently sampled groundwater ranged from 0.20 to 1.98 TU with an average of 0.93 TU (Table 3). These results show that for all formations combined, the groundwater in the study area has less than 2 TU. These values are identical to those measured by [9] in the lower sandstones of the Taoudéni basin in 2003. Based on the average tritium content of the rainfall water (5 TU) determined by authors of [30] in 2010 in the region, this suggests low water recharge in the Tamassari basin. The variability of tritium contents in the sampled waters indicates a different recharge of rainwater in the study area. It is also noted that the sedimentary formations (F22, F1, F2 and F15) have higher tritium contents than the basement formations (F10, F13, F17 and F7) (Table 3). This suggests that the sedimentary formations have a higher turnover rate than the basement formations.

**Table 3.** Tritium contents in groundwater in the basin.

| Long | Lat | Code | Tritium Content (TU) |
|---|---|---|---|
| 05′17′1″7″ W | 10′36′4″9″ -N | F13 | 0.64 |
| 05′18′1″6″ W | 10′37′3″3″ -N | F10 | 0.20 |
| 05′19′0″7″ W | 10′39′4″7″ -N | F2 | 1.35 |
| 05′20′3″6″ W | 10′36′1″9″ -N | F15 | 1.98 |
| 05′18′2″5″ W | 10′36′0″5″ -N | F17 | 0.63 |
| 05′18′4″5″ W | 10′34′3″7″ -N | F7 | 0.46 |
| 05′16′5″3″ W | 10′45′5″8″ -N | F22 | 1.30 |
| 05′19′0″8″ W | 10′45′1″0″ -N | F1 | 0.88 |

4.1.2. Chloride Contents of Groundwater

Chlorides are present in large quantities in seawater ($\pm19$ g/L). Their concentration in rainwater is about 3 mg/L. In groundwater, their concentration depends on the rocks crossed. In some areas, chloride inputs are associated with saline formations or anthropogenic sources such as chlorinated fertilisers, certain industrial activities and landfill leachate [46]. The chloride ions present in the analysed samples are in low concentrations, sometimes even at concentrations below the detection limit of the chemical analysis apparatus. Contents vary between 0.02 mg/L (sample F13, F15, F17, F7 and F1) and 3.50 mg/L (sample F2) with an average of 0.66 mg/L (Table 4). None of the samples showed a concentration higher than the World Health Organization's guideline value of 250 mg/L. This implies that there is no source of pollution and therefore the chloride ions present in the groundwater in the basin are only those from rainwater. The average content of chloride ions in rainwater determined during the high-water period (recharge period) in the Kompienga basin in Burkina Faso was 3.50 mg/L in 2005 [47]. From this result, the degree of groundwater recharge by rainwater can be assessed. Thus, only borehole F2 (3.50 mg/L), whose chloride content is approximately equal to that of rainwater, would present a significant recharge of the water table. Boreholes with low chloride contents (F10, F22), sometimes even below the detection limit (F13, F15, F17, F7 and F1), and would indicate very low or even non-existent recharge. In view of these results, it can be said that the recharge zones are not discriminated in the Tamassari basin, as was the case with the tritium contents.

**Table 4.** Chloride contents in groundwater in the basin.

| Long | Lat | Code | Chloride Content (mg/L) |
|---|---|---|---|
| 05′17′1″7″ W | 10′36′4″9″ -N | F13 | 0.02 |
| 05′18′1″6″ W | 10′37′3″3″ -N | F10 | 1.01 |
| 05′19′0″7″ W | 10′39′4″7″ -N | F2 | 3.50 |
| 05′20′3″6″ W | 10′36′1″9″ -N | F15 | 0.02 |
| 05′18′2″5″ W | 10′36′0″5″ -N | F17 | 0.02 |
| 05′18′4″5″ W | 10′34′3″7″ -N | F7 | 0.02 |
| 05′16′5″3″ W | 10′45′5″8″ -N | F22 | 0.72 |
| 05′19′0″8″ W | 10′45′1″0″ -N | F1 | 0.02 |

*4.2. Spatial Modelling Recharge Factors*

4.2.1. Recharge Factors Mapping

1.    Lithology and lineament density mapping

In the combined image of spectral bands 7, 2 and 1 (Landsat 5 TM images), the litho-logical units of the basin are well differentiated. This image is of excellent quality and facilitates the digitization of these lithological units. Overall, four lithological units can be distinguished in the basin: sandstone, gneiss, andesite and dolerite veins (Figure 6). In the image obtained by the different treatments (PCA1, 0° directional filter with 3 × 3 window), the lineaments stand out very clearly and can be mapped accurately. Indeed, the filters improve the perception of lineaments, corresponding to lithological or structural discon-tinuities, by causing an optical shadow effect on the image [33]. Thus, the calculation of the cumulative length of lineaments on 3 × 3 km$^2$ grids shows lineament density values ranging from 0 to 0.07 km$^{-1}$ (Figure 7).

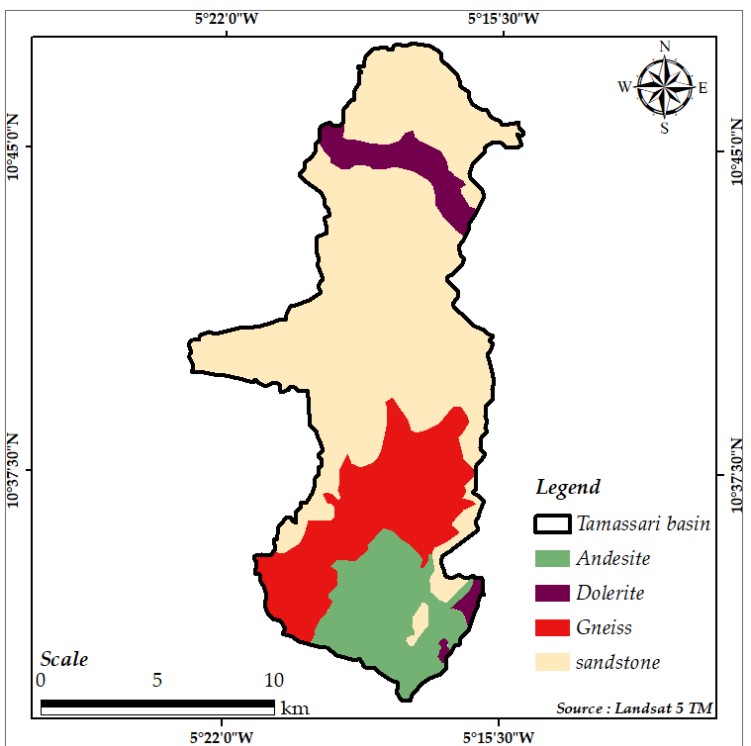

**Figure 6.** Lithology mapping.

2.    Land use mapping

Seven (07) land use units were mapped from the combination of spectral bands 5, 3 and 2 of the Landsat 8 OLI images. These are crops, rocky outcrops, gallery forest, bare soils, wooded savannah, bodies of water and marshy areas (Figure 8). In addition to the classes of variable land use from one season to another due to vegetation and bare soil variations, we considered the rocky outcrops as a formation that influences water infiltration rates. The confusion matrix of the support vector machines (SVM) algorithm showed the best classification with a kappa coefficient of 0.934 (Table 5), which has to be higher than 80% to validate the classification process. The values of kappa found with the other algorithms, namely the Gaussian mixture model (GMM), K-nearest neighbours (KNN) and random forest (RF), returned kappa coefficients of 0.929, 0.933 and 0.920, respectively. We observe that the differences in the kappa values of these different algorithms are not significant; this is perhaps due to the fact that the number of classes to be discriminated is low (seven classes in this study). According to [35], the number of classes influences the value of the coefficient and reducing the number of classes increases the value of the kappa.

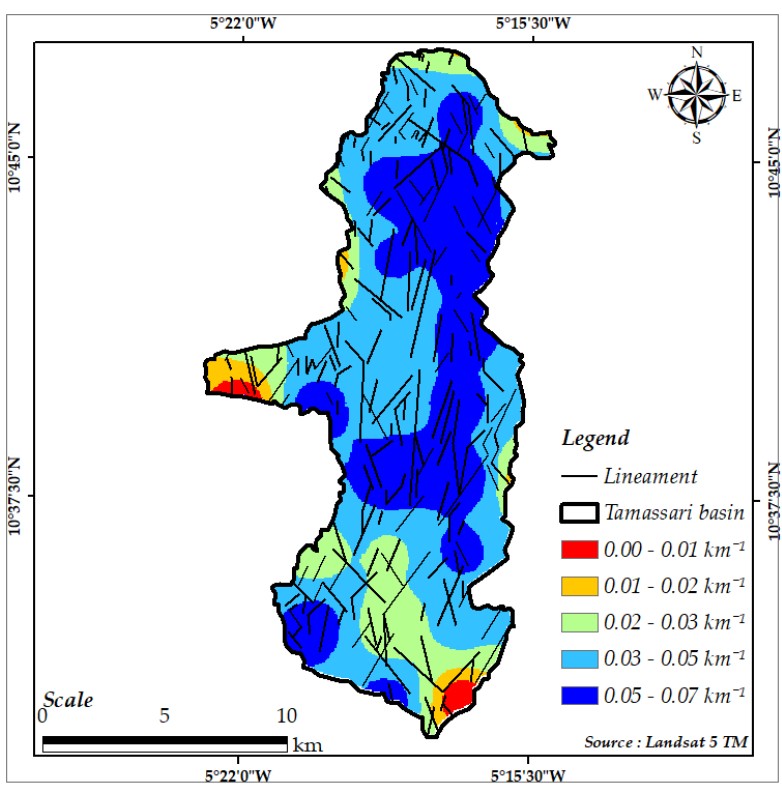

**Figure 7.** Lineament density mapping.

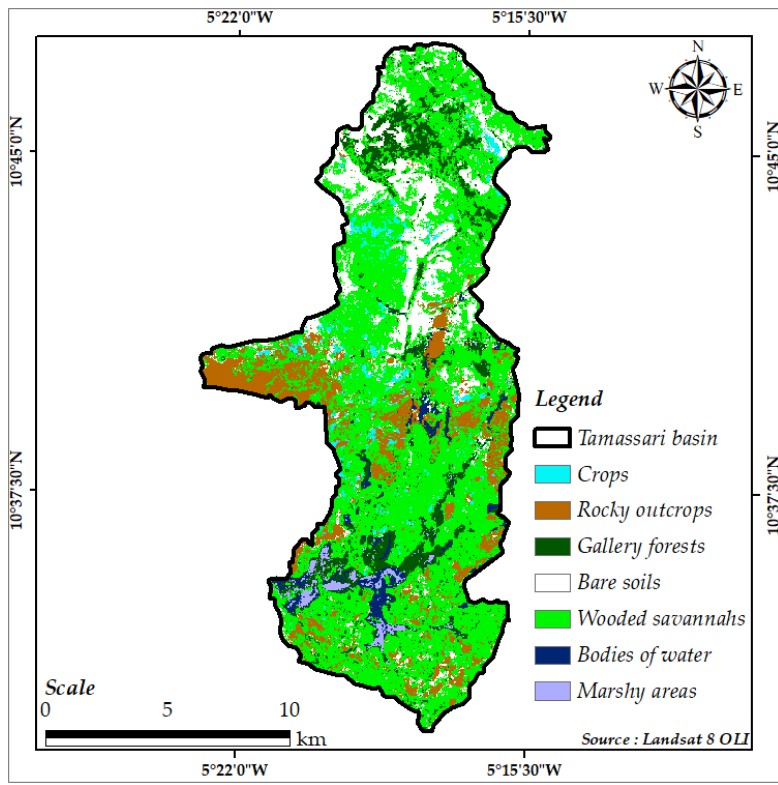

**Figure 8.** Land use mapping.

**Table 5.** Confusion matrix (Kappa = 0.934).

| | | Classified Image (SVM) | | | | | | | |
|---|---|---|---|---|---|---|---|---|---|
| | Class | Crops | Rocky Outcrops | Gallery Forests | Bare Soils | Wooded Savannahs | Bodies of Water | Marshy Areas | Total |
| Test areas | Crops | **34** | 0 | 0 | 0 | 1 | 0 | 0 | 35 |
| | Rocky outcrops | 0 | **247** | 0 | 0 | 1 | 0 | 0 | 248 |
| | Gallery forests | 0 | 0 | **67** | 0 | 23 | 0 | 0 | 90 |
| | Bare soils | 0 | 0 | 0 | **122** | 5 | 0 | 0 | 127 |
| | Wooded savannas | 3 | 0 | 7 | 1 | **479** | 2 | 0 | 492 |
| | Bodies of water | 0 | 0 | 0 | 0 | 6 | **40** | 0 | 46 |
| | Marshy areas | 0 | 0 | 0 | 0 | 0 | 1 | **44** | 45 |
| | Total | 37 | 247 | 74 | 123 | 515 | 43 | 44 | **1083** |

3.   Hydrographic network density and slope mapping

Data from the ALOS PALSAR DEM revealed important information about the density of the hydrographic networks and the slopes of the study area. The combination of these two factors provides an idea of the direction of the water flows and their distribution in the basin. The values for the density of the hydrographic network vary from 0 to 4 km$^{-1}$ (Figure 9). The slope map shows that the slopes of the study area vary from 0 to 65% (Figure 10).

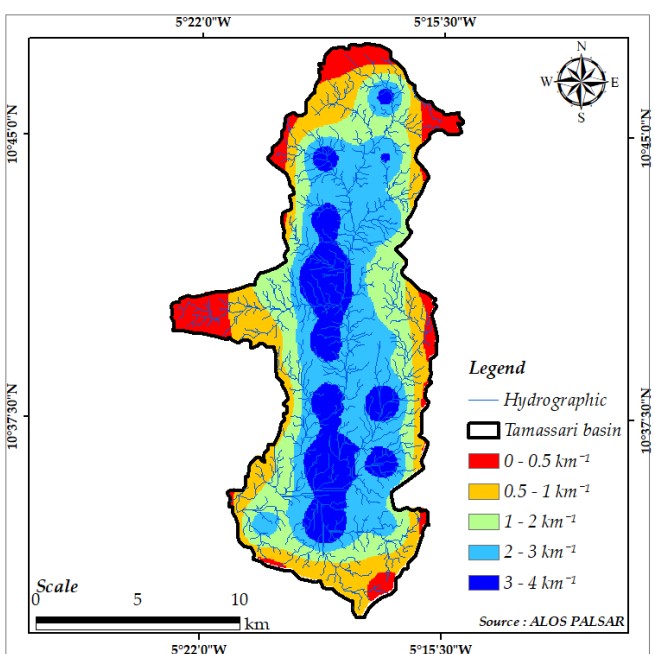

**Figure 9.** Hydrographic network density mapping.

4.   Soil type mapping

The soil data, as mentioned above (3.2.1.IV), shows four (4) soil types in the basin: fersiallitic soils; hydromorphic soils; poorly developed soils; and tropical ferruginous soils (Figure 11). The dominant soils are ferruginous soils with little leaching and leaching on sandy, sandy clay and clayey materials. Ferralitic soils on sandy clay are also found in the south and strips of poorly evolved soils on gravel in the central and northern parts of the basin. Hydromorphic soils are poorly represented in the north-eastern part of the basin [21]. These soils are generally poor in fertilising elements, especially nitrogen and phosphorus, with a poor structure. Their physical and hydrodynamic properties are therefore unfavourable to infiltration and water retention [22].

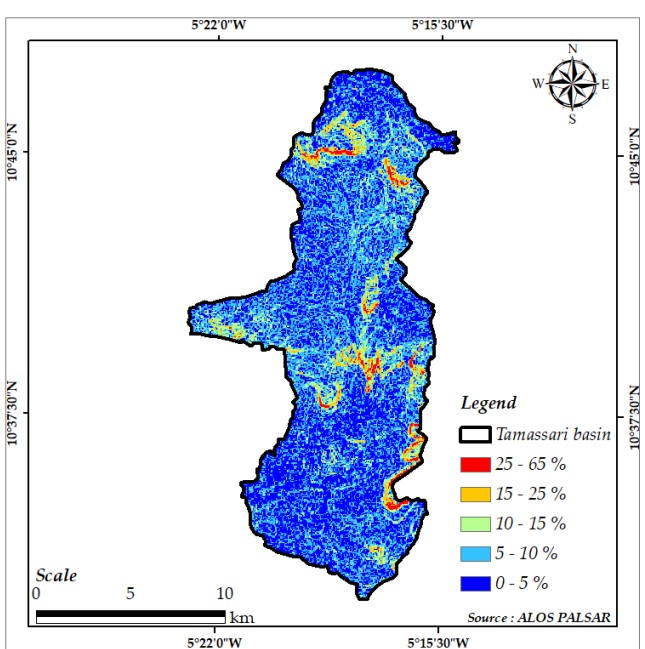

**Figure 10.** Slope mapping.

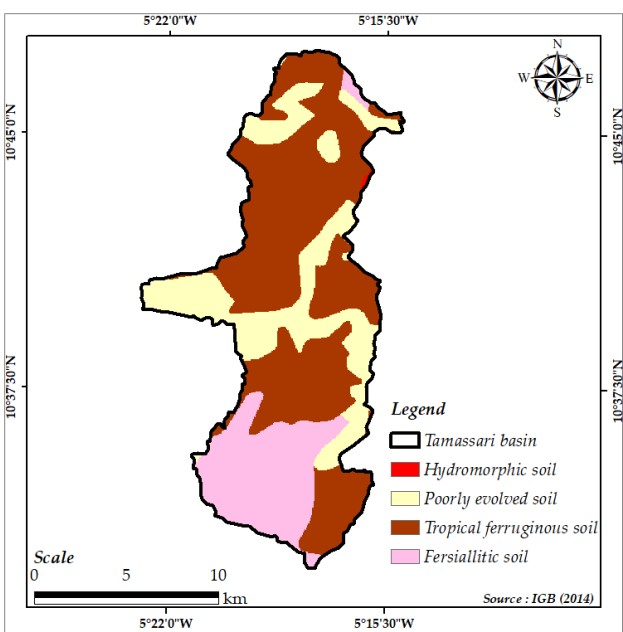

**Figure 11.** Soil type mapping.

4.2.2. Validation of Thematic Maps

The data obtained from the field work allowed us to understand the satellite images, and the truth points to validate the different thematic maps corresponding to recharge factors. As an example, Figure 12 illustrates the microstructure recognition activities on the outcrops identified in the study area. The location of these outcrops is shown on the colour composite of bands 7, 2 and 1 of the Landsat 5 TM images (Figure 12a). Microstructural measurements carried out directly on the sandstones Identified in the Kangoura locality to the south of the basin revealed fracture directions of N 30°, N 70°, N 130° and N 170° (Figure 12b). Fractures observed in the dolerite vein in the Diali locality in the north of the basin showed orientations of N 20°, N 60°, N 125°, N 150° and N 155° (Figure 12c). Structural measurements on the gneisses in the Kaguina locality in the centre of the basin revealed directions of N 80°, N 128°, N 140° and N 170° (Figure 12d).

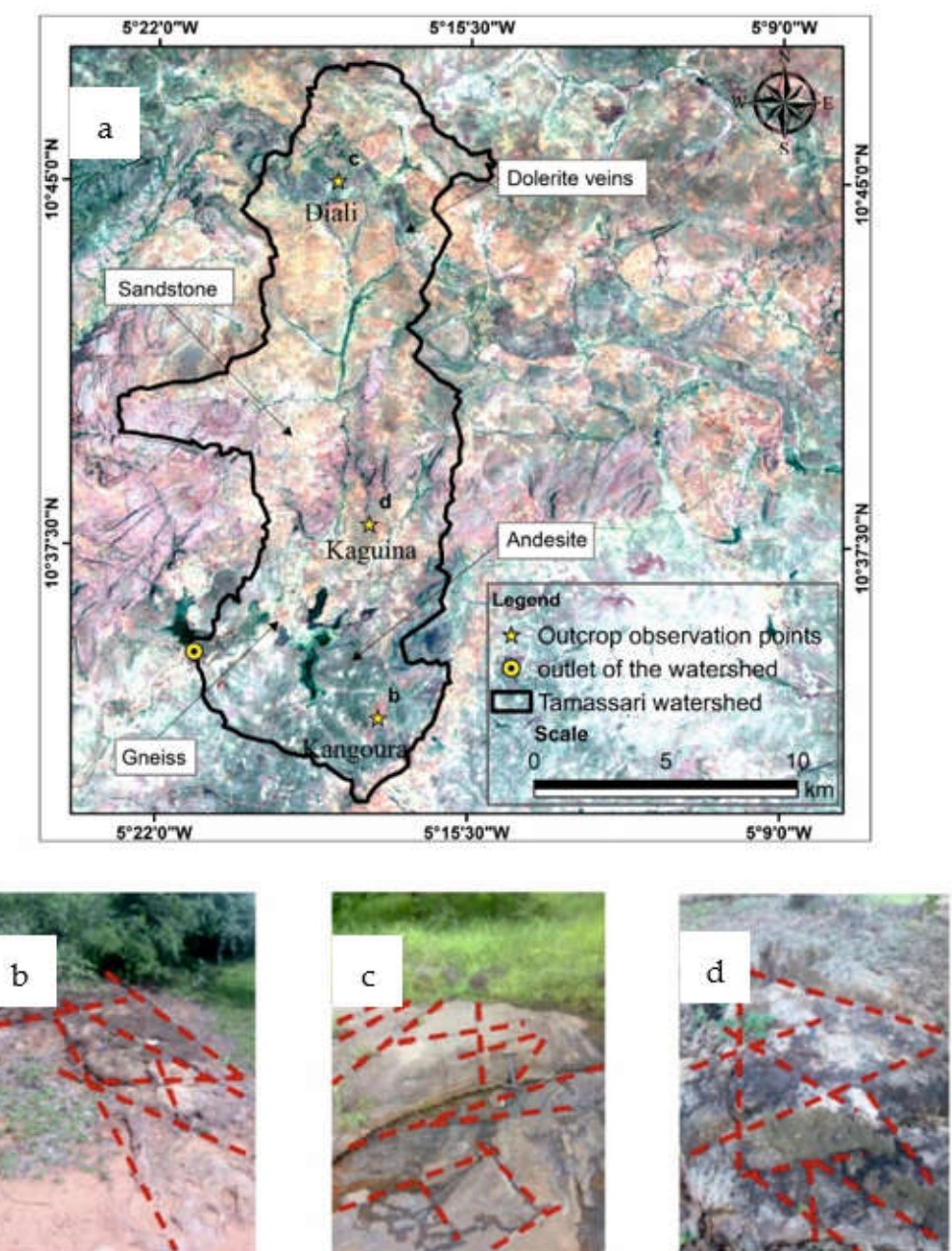

**Figure 12.** (**a**) Location of the outcrops on the colour composite of bands 7, 2 and 1 of the Landsat 5 TM images; (**b**) Kangoura fractured sandstone; (**c**) Diali fractured dolerite; (**d**) Kaguina fractured gneiss.

On the other hand, we were not able to study the andesitic rocks due to the absence of outcrops, probably because of a very high cover of detrital deposits in the area. However, all outcrop observations made were sufficient to validate the lithostructural map.

On the rosette, trends in lineament directions mapped from satellite imagery are primarily N0–60 (NE–SW) and N120–180 (NW–SE) (Figure 13).

The comparison of these lineament direction measurements with the structural measurements on outcrop shows a similarity. Indeed, the directions of the structural measurements made on the outcrops have been recognised on the directional rosette of the lineaments. In addition to structural measurements, hydrological lineaments were observed using GPS control points throughout the basin and correspond to either gullies, gallery forests or hydrographic networks (Figure 14).

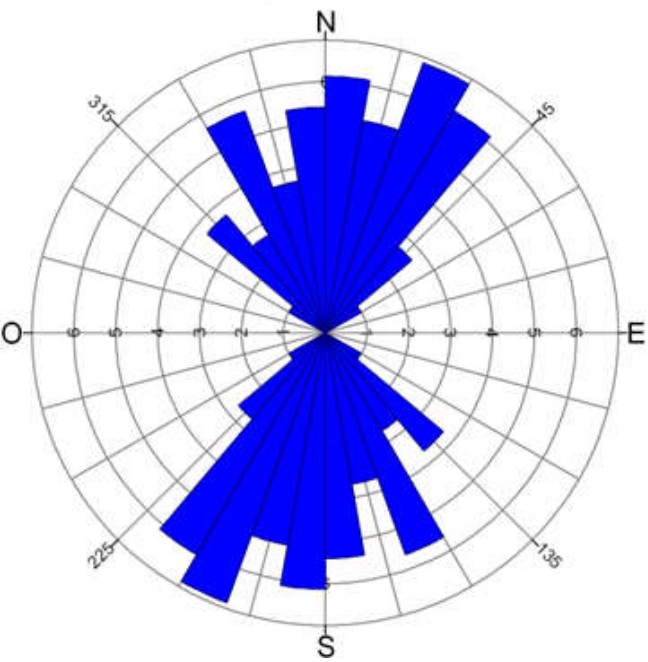

**Figure 13.** Directional rosette of the lineaments.

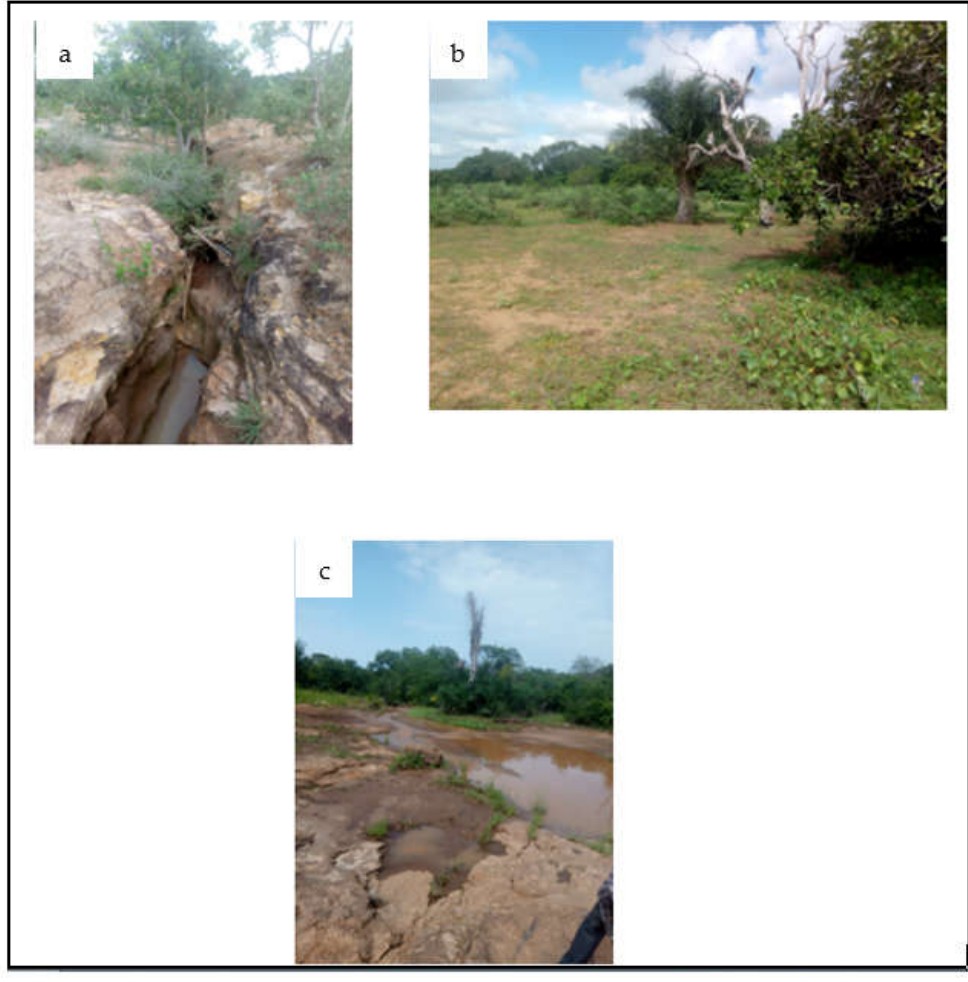

**Figure 14.** (**a**) View of a gully in the sandstone; (**b**) view of a gallery forest in the sandstone; (**c**) view of a hydrographic network in the sandstone.

### 4.3. Assignment of Ratings and Weights Calculation of Recharge Factors

The evaluation of the data for the different recharge factors allowed the assignment standardized ratings from 1 to 10 to all factor data according to their degree of involvement in recharge. This assessment is based on the hydrodynamic characteristics of the factor data determined by previous work [22,38,39]. The flow rate of an aquifer provides an idea of the degree of fracturing in the rock. Fractures are the preferred locations in the rock through which water percolates to the water table [38]. Thus, the frequency of fractures in the rock is a favourable factor for water storage in aquifers, but this criterion should be considered secondary to the degree of fracture opening [38]. Soil texture (percentage of clay, silt and sand) plays an important role in soil permeability [22]. The permeability of a soil is greater the higher its macroporosity is [22]. The presence of exploitable groundwater within the rocks is due to fracturing or fissuring. Areas of very high fracturing density are the pole of potential reservoir existence, while areas of very low fracturing density are very unfavourable for reservoir presence [39]. Areas of high hydrographic network density are likely to represent impervious areas with high runoff [39]. Areas with low slopes can be considered as favourable for infiltration and therefore recharge. High slopes favour rapid runoff and drainage of meteoric water [39]. Impervious surfaces (buildings, roads, etc.) considerably delay the recharge process. Vegetation cover, on the other hand, improves recharge, as it favours the confinement of water in the soil, thus preventing direct evaporation [39]. The ratings of the different recharge factor data are shown in Table 6.

**Table 6.** Rating of the data for each factor.

| Parameters | Hydrodynamic Classification | Standardised Ratings |
|---|---|---|
| Lithology (LT) | | |
| Andesite | Low | 1 |
| Dolerite | Good | 7 |
| Gneiss | Moderate | 3 |
| Sandstone | Excellent | 10 |
| Lineament density (LD) | | |
| 0–0.01 km$^{-1}$ | Low | 1 |
| 0.01–0.02 km$^{-1}$ | Moderate | 3 |
| 0.02–0.03 km$^{-1}$ | Average | 5 |
| 0.03–0.05 km$^{-1}$ | Good | 7 |
| 0.05–0.07 km$^{-1}$ | Excellent | 10 |
| Hydrographic network density (HD) | | |
| 3–4 km$^{-1}$ | Low | 1 |
| 2–3 km$^{-1}$ | Moderate | 3 |
| 1–2 km$^{-1}$ | Average | 5 |
| 0.5–1 km$^{-1}$ | Good | 7 |
| 0–0.5 km$^{-1}$ | Excellent | 10 |
| Soil type (ST) | | |
| Hydromorphic soils | Low | 1 |
| Poorly developed | Moderate | 3 |
| Tropical ferruginous soils | Good | 7 |
| Fersiallitic soils | Excellent | 10 |

**Table 6.** *Cont.*

| Parameters | Hydrodynamic Classification | Standardised Ratings |
|---|---|---|
| | Slope (SP) | |
| 25–65% | Low | 1 |
| 15–25% | Moderate | 3 |
| 10–15% | Average | 5 |
| 5–10% | Good | 7 |
| 0–5% | Excellent | 10 |
| | Land use (LU) | |
| Bare soils | Low | 1 |
| Rocky outcrops | Low | 1 |
| Water body | Moderate | 3 |
| Marshy areas | Moderate | 3 |
| Wooded savannah | Average | 5 |
| Crops | Good | 7 |
| Gallery forest | Excellent | 10 |

Using the analytical hierarchy process (AHP), weights are calculated by performing a pairwise comparison of the factors. The matrix resulting from the pairwise comparison of the factors is shown in Table 7. In this matrix it can be read, for example, that land use is five (05) times more important than soil types and five (05) times more important than lithology. Thus, this matrix is used to calculate the eigenvector corresponding to the weights to be applied to each factor as well as the maximum eigenvalue and the coherence indices to check the coherence of the judgement made when comparing the factors two by two.

**Table 7.** Comparison matrix and factors weights.

| Parameters | LU | LD | HD | SP | ST | LT | Weights | Consistency Indices |
|---|---|---|---|---|---|---|---|---|
| LU | 1 | 1 | 3 | 3 | 5 | 5 | 0.298 | $\lambda_{max}$ = 6.295 |
| LD | 1 | 1 | 3 | 5 | 5 | 7 | 0.343 | $CI$ = 0.059 |
| HD | 1/3 | 1/3 | 1 | 1 | 3 | 5 | 0.134 | $CR$ = 0.048 |
| SP | 1/3 | 1/5 | 1 | 1 | 3 | 5 | 0.126 | |
| ST | 1/5 | 1/5 | 1/3 | 1/3 | 1 | 3 | 0.063 | |
| LT | 1/5 | 1/7 | 1/5 | 1/5 | 1/3 | 1 | 0.036 | |

Notes: land use (LU); lineament density (LD); hydrographic network density (HD); slope (SP); soil type (ST); lithology (LT).

The consistency ratio (CR) obtained is 0.048 or 4.8%. Since its value is less than 10%, the entire comparison matrix is considered consistent; otherwise, pairwise comparison has to be modified to fulfil this requirement.

*4.4. Potential Recharge Mapping*

The summary map (Figure 15) of the potential recharge of the Tamassari basin, defined by the five zones, reveals that the areas with very low recharge represent only 4.12% of the area of the basin studied and are in the form of small beaches scattered in places in the basement rock formations. These areas are more marked between the locality of Lera and Kangoura and correspond to bodies of water and marshy areas. The areas with low recharge cover 24.74% and are mainly located in the southern part of the area studied,

precisely in the vicinity of Tamassari, M'para and Loumana. They are also found in the northern part of the basin at the level of the dolerite sill, which crosses the sedimentary formations. Average recharge areas represent 32.47% of the basin. Soils and rocky outcrops seem to control these areas better. However, there is an extension of this area around the main hydrographic networks. They are found in both sedimentary and basement formations. The areas with average to high recharge cover 35.05% of the total surface of the basin and are essentially encountered in the sandstone formations of the Proterozoic era, whereas the high recharge representing only 3.61% of the total area are highly represented in the North of the basin in the localities of Diali and Kankalaba. These areas are also found to the northwest of the M'pogona village, and they are related to the main hydrographic networks associated with well-defined channels (fault, fracture, etc.).

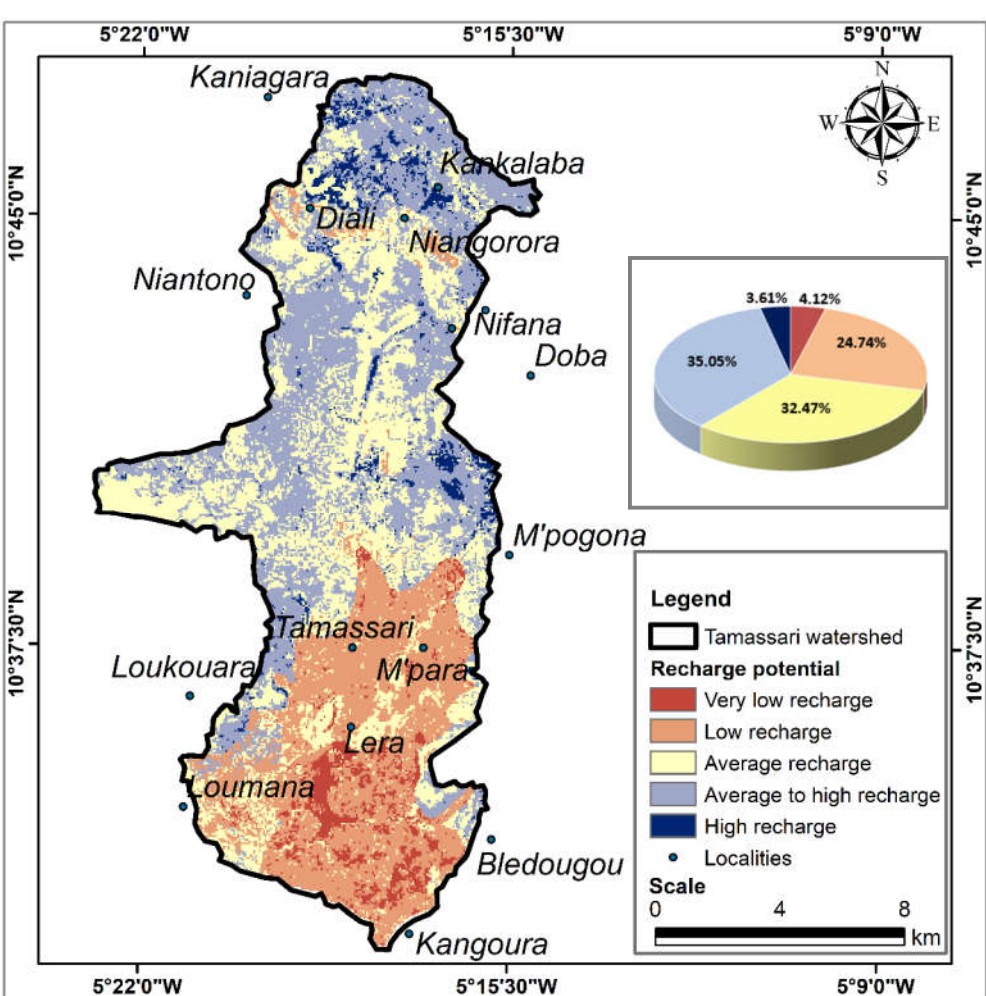

**Figure 15.** Recharge potential map of the Tamassari basin by the AHP method.

*4.5. Validation of the Potential Recharge Map*

The hydrochemical and isotopic analyses used in the assessment of the potential recharge were used to evaluate the modelized recharge based on the AHP method. The tritium and chloride contents obtained from the boreholes to the different recharge areas were identified by their corresponding classes on the recharge map. They are reported in the correspondence matrix illustrated in Table 8. The classification of tritium content was based on the current content of rainwater (5 TU), determined by authors of [30] in 2010. The closer the water content of the boreholes is to this value, the greater the recharge. Thus, according to this value, five (5) classes of recharge were defined: very low (0–0.20 TU), low (0.20–0.5 TU), average (0.5–1 TU), average-to-high (1–1.5 TU) and high (>1.5 TU). The same is true for chloride, which is currently present in rainwater at 3.5 mg/L [47]. Thus,

the closer the water content of the boreholes is to this value, the greater the renewal of the groundwater. According to this value, five (5) recharge classes have been defined: very low (0–0.02 mg/L), low (0.02–1 mg/L), average (1–1.5 mg/L), average-to-high (1.5–2 mg/L) and high (>2 mg/L). Tables 8 and 9 show the correlations of the two parameters with the recharge areas.

**Table 8.** Correspondence matrix between tritium contents and recharge areas.

| Tritium content (TU) | Recharge Areas | | | | |
|---|---|---|---|---|---|
| **Class** | **Very Low** | **Low** | **Average** | **Average to High** | **High** |
| Very low (0–0.20) | **F10** | | | | |
| Low (0.20–0.5) | | **F7** | | | |
| Average (0.5–1) | | | **F13, F17** | | **F1** |
| Average-to-high (1–1.5) | | | | **F2, F22** | |
| High (>1.5) | | | | | **F15** |

**Table 9.** Correspondance matrix between chloride contents and recharge areas.

| Chloride content (mg/L) | Recharge Areas | | | | |
|---|---|---|---|---|---|
| **Class** | **Very Low** | **Low** | **Average** | **Average to High** | **High** |
| Very low (0–0.02) | | **F7** | **F13, F17** | | **F1, F15** |
| Low (0.02–1) | | | | **F22** | |
| Average (1–1.5) | **F10** | | | | |
| Average to high (1.5–2) | | | | | |
| High (>2) | | | | **F2** | |

In Table 8, the different recharge zones are covered by the majority of boreholes with the corresponding tritium contents, with the exception of borehole F1, which is in the high recharge zone instead of the medium recharge zone. This result shows a very good correlation between the potential recharge modelized by the AHP method and the isotopic analysis. With regard to borehole F1, which is poorly classified, the phenomena of upward flow from deep aquifers could explain this discrepancy. Upward drainage of water is a very frequent phenomenon in arid zones [43]. Indeed, when water rises, it can cause mixtures between old and newly recharged water, leading to a decrease in tritium content. On the other hand, there is a poor distribution of chloride content in the recharge areas (Table 9). Most of the boreholes with very low chloride contents are distributed in all the recharge areas. This means that chloride contents are not reliable tracers in the study area when discriminating between different recharge areas.

## 5. Discussion

Remote sensing and GIS techniques proved their efficiency in the assessment of the recharge potential of aquifers in the study area. Five (5) classes of recharge degree were revealed. The very low degree of recharge observed in the localities of Léra and Kangoura is the result of the horizontal concentration of water near the surface in the absence of well-defined channels [48]. However, it is important to note that the degree of low recharge obtained is related to rainfall without the contribution of runoff and runoff concentrations. In fact, in artificial and natural water reservoirs (ponds), knowledge of the ability of the soil to recharge the water table is important. In the event of major floods, the height of the water concentrated in the reservoir should be assimilated to the local rainfall. This will advantageously increase the groundwater recharge potential compared to rainfall only

as a source of water input. If this water level is, for example, 3 m high, a contribution to recharge of 3.5 times the rainfall (for an average rainfall of 850 mm) could be expected. Thus, in the Sahelian strip, water retention sites are par excellence to those that recharge two to three times the amount of rainfall if human exfiltration and evapotranspiration are average [30,49]. Low recharge areas are related to basement rocks (gneiss, andesite). Although these formations are intensely affected by the Eburnean orogeny [24,50], the high thickness of weathering and the type of soil (clayey) that develops on these rocks would considerably reduce rainwater recharge. This observation is in agreement with the work of [14], showing that the clay texture of soils would prevent precipitation water from infiltrating. The average degree of recharge in soils and rock outcrops may be the result of a deficit in soil moisture in these land use units. Where the soil surface is indurated (rock, soil), runoff is high, and the soil surface takes time to become saturated. This means that the transfer of rainwater to depth is not fast enough. The degree of average to high recharge could be related to the geological nature of the sedimentary formations. These formations are made up of fine to medium quartzite to past conglomeratic in places. This would promote good recharge by percolation of precipitation water towards the water table. High recharge areas result from the rapid and deep recharge of rainwater through preferential pathways such as fractures or fissures and along major rivers. This fracture drainage phenomenon plays a fundamental role in water transport within fractured aquifers [51].

Overall, the medium and high recharge zones characterize the sedimentary formations of the Tamassari basin (Taoudéni basin) while the very low and low recharge zones occupy the basement domain. This shows that the Taoudéni sedimentary basin has a recharge potential with respect to the basement rock. This could be justified by the fact that the sandstone aquifer system is both porous and fractured, unlike the basement rock aquifer, whose recharge depends strongly on the presence of open fractures. These results go in the same direction as the work of [52], stipulating that the Taoudéni basin constitutes a powerful sandstone aquifer system. In addition, the temporal analysis of the piezometry over the entire Taoudéni basin revealed a general trend towards the rise in the piezometric level [53]. This result further confirms the good recharge areas mapped in the Tamassari basin. Indeed, the high piezometric levels in unconfined aquifers are due to the recharge of the water arriving there during rainfall events [30].

The good correlation between tritium contents and the different recharge areas shows the importance of using tritium to study groundwater recharge. Indeed, the presence of tritium in water indicates a current recharge. This isotope has been used efficiently in many studies to characterise recharge. We can mention the work of [9] in 2003 in the western part of Burkina Faso, where tritium contents were highlighted to characterise groundwater recharge. The results indicated three types of water: a first group of waters with a tritium content < 2 TU, i.e., old; a second group of waters with a content of 2 to 4 TU corresponding to a mixture of old and young waters and a third group of recent waters with a content of more than 4 TU. Similar studies were conducted by authors of [43] in 2012 to investigate the groundwater of the Dargol basin (Liptako–Niger). The results also showed several types of water. These results show that tritium is an excellent indicator of recent groundwater.

In contrast to chloride contents, there is no correlation with recharge areas. This could be related to the thick weathering and the type of soil (clay) that develops in the study area. Indeed, the fine clay layers retain the chlorides coming from the rainwater in the unsaturated area, thus preventing them from entering the groundwater. This would explain the fact that most boreholes show chloride contents below the detection limit (0.02 mg/L). Therefore, the use of chlorides for recharge studies is limited in the study area.

Thus, despite the fact that the number of boreholes does not cover the study area homogeneously, this study shows an overall trend in recharge, which is low for the basement formations and high for most of the sedimentary zone. This can be explained on the one hand by the favourable lithology (high permeability) of the sedimentary formations and on the other hand by the high density of fracturing observed in this sector, which is therefore favourable to infiltration.

This study has shown that remote sensing and GIS have enormous advantages in the spatialization of potential recharge areas in the Tamassari basin. Indeed, the use of a recent satellite image (Landsat 8 OLI of the year 2021) for land use mapping and corresponding to the date of groundwater abstraction, allowed an easy update of the recharge map. Furthermore, this study revealed that land use is the second most important factor (W = 0.298) in determining recharge areas. As for the significant influence of land use on the recharge, which has been the subject of certain studies [54,55], it must be approached according to a temporal approach covering the last 10 years [55] and based on remote sensing to produce land cover classes and produce recharge map with varying land use patterns.

Despite this importance, this factor is conditioned by the type of soil, which itself is dependent on the nature of the underlying rock. Moreover, the spatial database developed in the framework of this study can be used in the modelling of aquifers such as the hydrogeological model of groundwater flow in sedimentary aquifers using the HUF module of Modflow2000 [56] or any other model that can be interfaced with a GIS for a better follow-up by scientists as well as managers.

Mapping the potential recharge of aquifers using remote sensing and GIS techniques in Tamassari basin reveals some limits. The main difficulty lies in the definition of class limits and weights, which are assigned to the various factors entering into the realization of the GIS [15]. The choice of class limits is most often made according to the operator's ability to discern and his sense of judgment [57]. As is the case with methods based on the subjective appreciation of class limits, the final result of recharge mapping can be greatly influenced by the choices of these limits [12]. Therefore, a sensitivity study is required to determine the limits of the method and the influence of the choice of class limits on the result. The incorporation of probabilistic uncertainty into the AHP technique is considered a powerful approach to quantifying the sensitivity and uncertainty of the proposed model. The beta-PERT distribution has been widely used to model expert judgements and provide a close fit to normal distributions with little data [58,59]. This technique uses the most likely, minimum and maximum values of expert estimates to generate a probability distribution that measures the level of confidence in the AHP decision [60]. Moreover, the use of certain types of remote sensing data such as lineaments, which are considered as expressions of fractures and geological accidents is much controversial in the literature [15]. The number of lineaments observed generally depends on the methods of interpretation and the experience of the operator [15]. However, in our case, there is a good correlation between the lineaments and the geological features in the field.

## 6. Conclusions

The objective of this study was to determine the recharge potential of the aquifers of the Tamassari basin by spatial techniques and its validation using chloride ($Cl^-$) and tritium (3H) contents of groundwater. The methodological approach allowed reliable results to be obtained. Indeed, the validation of the thematic maps produced in this study by the field work ensured the reliability of the model input parameters. Similarly, the AHP method used allowed the relative importance of the factors to be assessed in relation to each other, leading to the determination of consistent weights for each factor. Moreover, the use of a recent satellite image (Landsat 8 OLI of the year 2021) for land use mapping and corresponding to the date of groundwater sampling, allowed an easy update of the recharge map. The novelty of this study is the use of a correlation matrix between the recharge areas identified by remote sensing and GIS techniques and the known tritium and chloride water contents of existing boreholes in the basin.

Thus, the recharge potential map obtained shows a generally good recharge trend for the sedimentary formations. The areas with low recharge are located in the basement zone. This map also allows for the identification of areas vulnerable to anthropogenic pollution, and therefore constitutes a decision support tool that should be made available to decisionmakers for a better spatiotemporal optimisation of the quantity and quality of

groundwater resources for the wellbeing of the surrounding population of the Tamassari basin. Furthermore, this study has shown the importance of using isotopes, especially tritium, to identify recharge areas in the region. In terms of perspectives, these results could be refined by a piezometric study by installing piezometers on the different recharge zones identified by the model.

**Author Contributions:** Conceptualization, I.K., H.C. and Y.K.; Methodology, I.K.; Validation, H.C., Y.K. and K.Z.; Formal analysis, I.K., H.C., Y.K. and K.Z.; Investigation, I.K.; Resources, K.Z.; Data curation, I.K.; Writing—original draft, I.K.; Visualization, I.K. and Y.K.; Supervision, H.C., Y.K. and K.Z.; Project administration, H.C. and Y.K.; Funding acquisition, H.C. and Y.K. All authors have read and agreed to the published version of the manuscript.

**Funding:** This research is part of the International Atomic Energy Agency (IAEA) TC RAF7019 research project, which aims to address the groundwater dimension of understanding and managing shared water resources in the Sahel region. The authors thank the International Atomic Energy Agency (IAEA) for the financial support to this research project and for the realisation of this study between Joseph KI-ZERBO university (Burkina Faso, home university) and Tunis El Manar university (Tunisia, host university), in collaboration with the University of Sfax (Tunisia). The authors would also like to thank the Programme d'Appui à l'Enseignement Supérieur (PAES) for its financial support for this research.

**Acknowledgments:** The authors would like to thank the staff of the Hydraulics and Environemental Modeling Laboratory (LMHE) of National School of Engineers of Tunis (ENIT) and of the Laboratory of Geosciences and Environment (LaGE) of University Joseph Ki-ZERBO (UJKZ) of Ouagadougou as well as the staff of Laboratory of Radio-Analysis and Environment (LRAE) of National School of Engineers of Sfax (ENIS) for all the efforts deployed during the course of this research work. We would also like to thank the editors and the anonymous reviewers. Their comments have significantly improved the manuscript.

**Conflicts of Interest:** The authors declare no conflict of interest.

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
