# Peer review of "Assessment of Aquifer Recharge Potential Using Remote Sensing, GIS and the Analytical Hierarchy Process (AHP) Combined with Hydrochemical and Isotope Data (Tamassari Basin, Burkina Faso)"

_water, doi:10.3390/w15040650_

Round 1

Reviewer 1 Report

General comments: 

This is an interesting analysis, bringing together remote sensing and other data sources in a logical manner.   The different data sets are not clearly documented and some of the data sources are old.   Questions:  What do the results imply for water management? Is it to identify areas for augmenting recharge?  How do you go to next step of quantifying recharge rates, going beyond the qualitative ranking of “good” versus “bad?”  It would be helpful to develop models to quantitatively test hypotheses and simulate different management scenarios.

Please note that I am not an expert in Landsat processing and analysis.   I presume that other reviewers bring that expertise. 

Line by line comments:

12-13, 34-6 –Provide more info on precipitation trends from reference # 2, Kabore et al, 2017.  That paper conveys more complexity within the overall decline in precipitation.   

43- what does “capitalization” refer to in this context?

98 – fig. 1.  Show scales on all maps.  Relationship of 3rd map to 2nd map is not clear.  Show more features on 2nd map. Label rivers on 3rd map.  Provide complete references for sources of this figure – not just Palos Palsar

110-11 “16.57% reduction in the class of cover 110 rate between 80 and 100,…” Here and elsewhere, including figure 2, define cover classes and explain what the units are.   

113-14 – show this on fig. 1.

133-5  “…between 1988 and 1992..” Have any more recent data?

139-45 - show this on a cross section, could be schematic. 

139 – explain what you mean by “flow…towards bedrock.”

177-8:  show these recent data.  Would be helpful to present hydrographs that show trends. 

183. depths of sampling intervals?

188 hydrogeologic

257-8 - include reference(s) for this.

286-7- “Thus, the higher the productivity….”  This assumption is very dependent on specifics of a given area.  E.g., how fractured. Granite could be very low.  Volcano sedimentary could be very high.  These generic values cannot be automatically applied.

341. – refer to figure 1 where the boreholes are shown.

357 – assumption of rainfall being only source of chlorides apparently does not hold true in the study area.

372-3. – “suggests that the sedimentary formations have a higher turnover rate than the basement formations,” This makes sense – what one would expect. 

438 – fig 8 correct figure captions- this is NOT lithology

440 fig 9, same as above re caption

452 - This is fig10 not fig. 6

488 - please elaborate on the discussion in these references.  This is important part of your assessment.

589-70 – see comment above for line 357.

576-80. – this is important assumption when interpreting results. 

652-3. – another important assumption.

Author Response

We would like to thank the anonymous reviewers. We appreciate all your comments. These comments have greatly improved the manuscript.

Reviewer 2 Report

Please explain why you determined the chlorides in the water for the assessment of the aquifer recharge zone, and not the microelements characteristic of the rocks that are in the zone from which the aquifer is recovered.

Another comment refers to the fact that the natural abundance of hydrogen isotopes  1H 2H and 3H (tritium) are 99.985%, 0.015%, and 10exp-18% respectively. Why did you use the measurement results of gaseous radioactive (decaying) tritium, which is naturally present in very small concentrations, which is why the measurement uncertainty of these results is very high, instead using the 2H isotopes and oxygen isotopes, which is the more commonly used way of estimating the source of aquifer recharge.

Author Response

(The authors gave the same response as above.)

Reviewer 3 Report

The manuscript titled “Assessment of aquifer recharge potential using remote sensing, GIS and the analytical hierarchy process (AHP) combined with hydrochemical and isotope data (Tamassari watershed, Burkina Faso)” is within the scope of the journal but I have doubt that it is in scope of the Special Issue which is “High-Resolution Monitoring and Modelling for Water Resources Management: New Sensors, New Approaches and Applications”.

The study tried to map the recharge potential of the existing aquifers in Burkina Faso, Africa making use of remote sensing and GIS techniques and to make a validation based on chloride and tritium contents in the borehole water. The study is very important from the perspective that study area is highly dependent on groundwater resources.

Study is conceptually sound. The novelty is highlighted. Methodology flowchart is expressing the methodology employed. Though there are minor corrections required to make it publishable in this journal. Following are the comments from my side:

1.      More references can be added to introduction. Authors may refer to these :-

i)       https://www.nature.com/articles/s41598-019-38567-x

ii)      https://link.springer.com/article/10.1007/s12524-019-01027-0

iii)    https://www.mdpi.com/2306-5338/9/12/224

iv)    Shah, T., Molden, D., Sakthivadivel, R., & Seckler, D. (2001). Global groundwater situation: Opportunities and challenges. Economic and Political Weekly, 4142-4150.

v)      https://www.tandfonline.com/doi/abs/10.1623/hysj.54.4.781

2.      The figure need to be made more uniform and legible. Use same font in all the figure. Try to improve the resolution. Also maps can be made more attaractive by removing unnecessary things in the maps (for eg. No need to write heading “Label” above label. Also no need to provide a separate box within the map for label).

3.      There are minor English errors in the manuscript. For eg. Line 416 “The values ok kappa found with the other algorithms, namely” of is witten as ok.

 l

Author Response

(The authors gave the same response as above.)

Reviewer 4 Report

The article "Assessment of aquifer recharge potential using remote sensing, GIS and the analytical hierarchy process (AHP) combined with hydrochemical and isotope data (Tamassari watershed, Burkina Faso)", written by four authors representing scientific institutions: University of Tunis El Manar, National School of Engineers of Tunis, Tunisia; University Joseph Ki-ZERBO of Ouagadougou, Unity formation and research, Burkina Faso; The University of Sfax, National School of Engineers of Sfax, Tunisia, is a case study and methodical.

 The purpose of this study was to assess aquifer recharge potential using remote sensing, GIS and the analytical hierarchy process (AHP) combined with hydrochemical and isotope data in the Tamassari catchment (Burkina Faso). The processing carried out on the Landsat 5 and Landsat 8 images combined with a digital elevation model (ALOS PALSAR), highlights the lithological, linear, and topographical characteristics of the study area. In addition, various supervised classification algorithms were used to produce the most accurate land use map. Field campaigns were conducted to validate the thematic maps resulting from the geospatial data processing and to collect water samples for hydrochemical (chloride) and isotopic analysis (tritium). The AHP method was used to derive recharge factor weights. The resulting recharge map showed an agreement between the recharge classes derived from spatial modeling and the tritium isotope analyses.

 Research results, discussion, using advanced research techniques, and conclusions are logical, and at a good scientific level. Among the critical comments / for discussion, the following should be mentioned: limited characteristics of the geological structure, of hydrological and hydrogeological conditions (including the lack of catchment water balance), the imprecise definition of land use (are "rocky outcrops" and "bare soils" forms of land use?) and its role in groundwater recharge; subjective definition of class boundaries and weights that are assigned to various factors included in the GIS; how the potential recharge of groundwater in particular parts of the catchment is quantified.

Detailed comments are provided below.

 Title of the article

1/ I suggest using the word "catchment" or "basin" instead of "watershed"; the note applies to the entire text

Abstract

1/ Explain the abbreviation - AHP method

2/ There is no information about when the investigations presented in the article were performed; the note applies to the entire text

2. Study area

1/ Figure 1. Location and geology of the study area. – the presentation of the geological structure of the study area is unclear. I see only one piece of information - "basement rock" – complete

2/ Figure 2. Evolution of vegetation cover classes between 2007 and 2014 ([20]). – write in what units the "Classes of vegetation cover" are characterized

3/ Figure 3 : Piezometric record between 1988 and 1992 of the lower sandstone, Taoudéni basin ([9]). – the research concerns the 20s of the 21st century, so why not present the current results of  observations? If groundwater is being characterized, it should be a lower sandstone aquifer (not lower sandstone). This remark applies to the entire text - we are not referring to geology but to aquifers

3. Materials and Methods

1/ 1 paragraph - six steps were distinguished, not five steps

 3.1. Data

1/ "... and water samples were taken from eight (8) boreholes during the high-water period in 2021" - low-precision characteristics, exact dates of water sampling should be provided

 3.2. Spatial Modelling of recharge

1/ "... image acquire in October 2021 images of October 2021" - is it not better to simplify the sentence.

 3.2.3.Hydrodynamic classification and data standardization

1/ Table 1. Influence of geology on the productivity of structures - the term "productivity" is not a strictly hydrogeological term, it should be changed; this also applies to text

2/ Table 1. Write what is the lithology of "Green rocks"

 4. Results

1/ Table 3. Tritium contents in groundwater in the watershed; Table 4. Chloride contents in groundwater in the watershed - use catchment/basin instead of "watershed"

2/ Table 6. Rating of the data for each factor. – only "Lithology" and "Lineament density" have ordered values of standardized ratings from 1 to 10. Standardized ratings for other parameters should be ordered in the same way.

 5. Discussion

1/ In the Discussion, more attention should be paid to land use

 6. Conclusions

1/ In the Conclusions, it can be emphasized to what extent the results of the presented research are representative and for what type of natural and hydrogeological environment, and in what climatic conditions the obtained research results can be used.

 References

All items listed in the References are cited in the text.

 The article is on a good scientific level, but it requires supplementation in the field of natural science, presentation of the methodological basis for defining class boundaries and weights that are assigned to various factors included in the GIS; introduction of quantitative data on the potential supply of groundwater in the studied catchment, as a result of the conducted research.

It should find a wide group of readers dealing with hydrogeological research, especially groundwater recharge with the use of modern research tools - remote sensing and GIS techniques, isotopic and hydrochemical research.

Author Response

(The authors gave the same response as above.)

Round 2

Reviewer 2 Report

The authors explained the  choice of experimental design, but still could not change the results of the measurement of the isotopic composition because that would require additional tests. Given that the isotopic composition is only a small part of the research that supports the hypothesis about the aquifer discharge potential, I accept this paper in its final version.